# Utilization of Intermediate Wheatgrass (*Thinopyrum intermedium*) as an Innovative Ingredient in Bread Making

**DOI:** 10.3390/foods12112109

**Published:** 2023-05-24

**Authors:** Buket Cetiner, Vladimir P. Shamanin, Zeynep H. Tekin-Cakmak, Inna V. Pototskaya, Filiz Koksel, Sergey S. Shepelev, Amanzhol N. Aydarov, Bayram Ozdemir, Alexey I. Morgounov, Hamit Koksel

**Affiliations:** 1Department of Quality and Technology, Field Crops Central Research Institute, Ankara 06170, Türkiye; buket.cetiner@tarimorman.gov.tr; 2Department of Agronomy, Breeding and Seed Production of the Agrotechnological Faculty, Omsk State Agrarian University, 1 Institutskaya pl., Omsk 644008, Russia; vp.shamanin@omgau.org (V.P.S.); iv.pototskaya@omgau.org (I.V.P.); ss.shepelev@omgau.org (S.S.S.); an.aydarov@omgau.org (A.N.A.); 3Department of Food Engineering, Faculty of Chemical and Metallurgical Engineering, Davutpasa Campus, Yildiz Technical University, Istanbul 34349, Türkiye; hazal.cakmak@yildiz.edu.tr; 4Department of Nutrition and Dietetics, Health Sciences Faculty, Istinye University, İstanbul 34010, Türkiye; 5Department of Food and Human Nutritional Sciences, University of Manitoba, Winnipeg, MB R3T 2N2, Canada; filiz.koksel@umanitoba.ca; 6Department of Plant and Genetics, Field Crops Central Research Institute, Ankara 06170, Türkiye; bayram.ozdemir@tarimorman.gov.tr; 7Saudi Arabia Country Office, Food and Agriculture Organization of the United Nations, Riyadh 11421, Saudi Arabia; alexey.morgounov@gmail.com

**Keywords:** intermediate wheatgrass, bread, sustainability, baking, loaf volume, yellow pigment contents, antioxidants

## Abstract

Intermediate wheatgrass (IWG; *Thinopyrum intermedium*), a nutritionally dense and sustainable crop, is a promising novel ingredient in bakery applications. The main aim of this study was to investigate the potential of IWG as a novel ingredient in breadmaking. The second aim was to investigate the characteristics of breads substituted with 15, 30, 45, and 60% IWG flour compared to control bread produced using wheat flour. The gluten content and quality, bread quality, bread staling, yellow pigment, and phenolic and antioxidant properties were determined. Enrichment with IWG flours significantly affected the gluten content and quality and bread characteristics. Increased levels of IWG flour substitution significantly decreased the Zeleny sedimentation and gluten index values and increased the dry and wet gluten contents. The bread yellow pigment content and crumb b* colour value increased with the increasing level of IWG supplementation. IWG addition also had a positive effect on the phenolic and antioxidant properties. Bread with 15% IWG substitution had the highest bread volume (485 mL) and lowest firmness values (654 g-force; g-f) compared to the other breads, including the control (i.e., wheat flour bread). The results indicated that IWG has great potential to be used in bread production as a novel, healthy, and sustainable ingredient.

## 1. Introduction

According to the Food and Agriculture Organization of the United Nations, improving efficiency in the use of resources is crucial to sustainable agriculture and a food-secure world [1]. Since perennial plants have longer growing seasons and continuous ground cover than annual plants, they have been suggested as more sustainable alternatives to annual plants [2]. Perennials remain in the soil year-round and provide extensive root systems capable of holding water and nutrients [3]. As such, perennial crops instead of annual crops can potentially open doors to a new era of agriculture that is more resilient to climate change, capable of soil carbon sequestration, and environmentally friendly [4]. Although more work is needed, perennial grains are compatible with current interests and policies including food security, soil health, water quality, farmer interest in innovative practices, and sustainable cropping systems [5].

Perennial wheat presents a promising novel strategy to increase food production and diversify agroecosystems in light of the aforementioned concerns [5,6]. Intermediate wheatgrass (*Thinopyrum intermedium*, IWG) is one of the wild perennial grass relatives of wheat. It has been traditionally used in wheat breeding for obtaining wheat–wheatgrass hybrids and varieties with introgressions of new genes for economically valuable traits. The direct domestication of *Th. intermedium* is an alternative seed production approach. For example, as a result of many years of domestication work at The Land Institute (Salina, KS, USA), the wheatgrass variety Kernza was developed and used for seed production, green mass, and hay [4,7]. Similarly, at Omsk State Agrarian University (Russia), the variety Sova was developed by a mass selection of the most winter-hardy biotypes, with their subsequent combination from the population of wheatgrass obtained from The Land Institute. The variety Sova has an average grain yield of approximately 9 c/ha, green mass yield of 210 c/ha, and hay yield of 71 c/ha [8,9].

One of the most popular staple foods around the globe is bread, a wheat-based product. IWG has great potential to be incorporated as a novel ingredient in bread, due to its higher protein and dietary fibre content when compared to bread wheat [10]. In addition, IWG was reported to contain relatively higher levels of certain hydroxycinnamic acids, notably, ferulic acid, when compared to wheat [3]. However, IWG flour’s ability to form a gluten network during dough mixing is low, producing relatively less-extensible doughs, and overall poorer breadmaking performance and loaf quality when compared to wheat [10].

Interest towards the use of IWG as a novel ingredient in foods has been increasing, especially in the last fifteen years [11]. However, there has been no systematic investigation of the effects of IWG on gluten properties as well as dough and bread quality characteristics. This study addresses these research gaps by investigating the potential of IWG as a novel ingredient in breadmaking by analysing the properties of fortified breads produced by replacing the white flour in bread formula with IWG flour. The specific objectives were to (1) assess the gluten properties and dough mixing characteristics of IWG-incorporated flours, (2) evaluate the loaf quality and staling properties of breads made from these flours, and (3) analyse the yellow pigment content, total phenolic content, and antioxidant activity of these breads. Finally, all IWG-incorporated flour, dough, and bread properties were compared to those of hard red winter wheat flour, dough, and bread.

## 2. Materials and Methods

### 2.1. Materials

Bread wheat (*cv*. Cavus, *Triticum aestivum*) and intermediate wheatgrass (*cv*. Sova, *Thinopyrum intermedium*) were used in the study. Cavus was obtained from the Field Crops Central Research Institute while Sova was obtained from Omsk State Agrarian University. Cavus is a hard red winter wheat with very strong gluten properties.

The bread wheat (*cv*. Cavus) and intermediate wheatgrass samples were milled according to the American Association of Cereal Chemists International (AACCI) Method 26-70.01 [12] using a Chopin CD 1 Laboratory Mill (Chopin Technologies, Villeneuve La Garenne, France). Flours were allowed to rest for 2 weeks prior to use. The chemicals used in the study were of analytical grade unless stated otherwise.

### 2.2. Methods

#### 2.2.1. Quality Evaluation of Flour

The protein contents of the flours were determined by combustion nitrogen analysis (Leco FP828, St. Joseph, MI, USA) calibrated with EDTA according to the AACCI Method 46-30 [12]. Correction factors of 5.7 and 6.25 were used for bread wheat and IWG flour, respectively. The Zeleny sedimentation values of the flours were obtained according to the ICC (International Association for Cereal Science and Technology) Method 116/1 [13]. Wet and dry gluten and gluten index (Perten Glutomatic and Glutork 2020, Huddinge, Sweden) of the flours were determined according to the AACCI Method 38-12.02 [12]. Moisture content was determined using the AACCI Method 44-15A [12]. Colour L*, a*, and b* values of flour samples were measured using a colour-view spectrophotometer (Gardner BYK, Columbia, MD, USA) according to Method E 1164 [14].

#### 2.2.2. Bread Making and Quality Evaluation

Farinograph properties of the doughs were determined according to the AACCI Method 54-21 [12] using a Farinograph (Brabender Farinograph-AT, Duisburg, Germany) equipped with a 50 g bowl. From the Farinograph curves, the dough development time (min), water absorption (14% moisture basis), stability (min), softening degree (BU, 12 min after the development time), and Farinograph quality number were obtained.

The formulation made only from wheat flour was defined as control. Four sets of bread were baked, replacing 15%, 30%, 45%, and 60% of the wheat flour with IWG flour, respectively. Breads were produced using the modified AACCI Method 10-10B [12]. The bread formula contains flour (100 g, 14% mb), salt solution (non-iodized, 25 mL, 6.0%), yeast suspension (25 mL, 8.0%), and water (determined by the water absorption value obtained from the Farinograph curves). A pin mixer (National Mfg, Lincoln, NE, USA) was used to mix the doughs, followed by fermentation. After 30 min of fermentation, the dough samples were punched and another 30 min of fermentation was allowed. The dough was shaped and placed in pans after the second fermentation. The final proof took 55 min. The loaves were baked for 25 min at 230 °C in a laboratory rotating oven (Despatch, Minneapolis, MN, USA). The baking tests were performed in duplicate.

After cooling the loaves at room temperature for 2 h, the bread volume was determined by a rapeseed displacement method according to AACCI Method 10.05-01 [12] using a loaf volumeter (National Mfg, Lincoln, NE, USA). Breads were placed in plastic bags and stored for 1 and 3 days at room temperature for the determination of crumb firmness. Firmness of breads was determined according to AACCI Method 74-09 [12]. For texture analysis, Stable Microsystems TA-XT plus texture analyser (Godalming, Surrey, England) with a 36 mm cylinder probe and a 50 kg load cell was employed. At a test speed of 1.7 mm/sec, the force (firmness, g) necessary to compress two slices each with a thickness of 1.25 cm by 40% was calculated. Colour L*, a*, and b* (D65, 10°) values of bread samples were measured using a spectrophotometer (Miniscan by HunterLab, Reston, VA, USA) according to Method E 1164 [14].

Quality characteristics (symmetry, crust colour, crumb cell structure, crumb colour, and softness) of the bread samples were evaluated by a panel of three experts experienced in bread quality evaluation. Bread samples were provided in duplicate under normal white light at room temperature. The symmetry and crust colour of the breads were scored using the 5-point numerical rating scale (1: poor and 5: very good), while the crumb cell structure, crumb colour, and softness of the breads were scored using the 10-point numerical rating scale (1: poor and 10: very good). The final score was reported as the mean of scores determined by all panellists.

#### 2.2.3. Total Phenolics, Antioxidant Activity, and Yellow Pigment Content Measurements

Bread slices (1.5 cm thickness) were dried in an oven (40 °C) for 24 h, then ground and sieved through a 35-mesh screen. The ground and sieved sample was mixed with a methanol/distilled water solution (80:20) at a ratio of 1:10. Then, the mixture was incubated by shaking at 200 rpm at 25 °C for 12 h in the dark. After the incubation, the mixture was centrifuged at 2800× *g* for 15 min (Nüve, NF 800R, Ankara, Türkiye). The supernatant, i.e., the bread extract, was filtered with Whatman no. 1 filter paper.

The total phenolic contents (TPCs) of the samples were determined spectrophotometrically according to the modified method reported by Blainski et al. [15]. Briefly, each of bread extracts (0.5 mL) was mixed with 2.5 mL of Folin Ciocelteau’s phenol reagent (tenfold diluted with distilled water). Then, 2 mL of Na_2_CO_3_ solution (7.5%, prepared with distilled water) was added to this mixture. After 30 min incubation in dark at 25 °C, the absorbance value was read at 765 nm using a UV/VIS spectrophotometer (Shimadzu UV-1800, Kyoto, Japan). Gallic acid was used as a standard to establish the calibration curve. TPCs of samples were expressed as mg gallic acid equivalent (GAE) per 100 g of bread (mg GAE/100 g bread).

The 1,1-diphenyl-2-picrylhydrazyl (DPPH) radical scavenging activity of the bread extracts was determined according to the method of Kedare and Singh [16]. Following this method, the bread extract (100 μL) was mixed with 3.9 mL of freshly prepared DPPH radical solution (3.9 mg DPPH/100 mL methanol). After 60 min incubation in the dark at 25 °C, the absorbance value was measured at 515 nm using a spectrophotometer (Shimadzu UV-1800, Kyoto, Japan). The results were expressed as mg Trolox equivalent per 100 g bread (mg TE/100 g bread).

The ABTS radical–cation scavenging capacity of the bread extracts was measured following the modified method described by Opitz et al. [17]. In the dark, a mixture of ABTS solution and bread extract (2 mL of ABTS solution and 100 µL of bread extract) was prepared and incubated at 25 °C for 6 min. After incubation, the absorbance value was measured at 734 nm using a spectrophotometer (Shimadzu UV-1800, Kyoto, Japan). The results were expressed as mg Trolox equivalent per 100 g bread (mg TE/100 g bread).

Yellow pigment content of the flours and breads was determined in terms of beta-carotene equivalent according to the “determination of pigments” method as described by the AACCI Method 14-50.01 [12].

#### 2.2.4. Statistical Analysis

All experiments were performed in duplicate, and the mean values were recorded. Results were analysed using a one-way analysis of variance (ANOVA) using the software JMP (Version 13.2.1, SAS Institute Inc., 2013, Cary, NC, USA). When significant (*p* < 0.05) differences were found, the least significant difference (LSD) and t-test were used to determine the differences among means.

## 3. Results and Discussion

### 3.1. Quality Evaluation of Flours

The protein content of the IWG (N × 6.25) and control (N × 5.70) flour were 19.3% and 12.0%, respectively, on a dry-matter basis (Table 1). IWG flour’s protein content is substantially higher than the typical protein content of common bread wheats. Pototskaya et al. [9] stated that due to its relatively smaller-sized seeds, IWG contains significantly less starch (46.7%) compared to wheat (72%), which translates to a relatively higher protein concentration for IWG seeds and flour. Gazza et al. [18] determined the protein content of four perennial wheat derivatives including *Thinopyrum intermedium* as a parent and three commercial wheat cultivars. They reported that the perennial wheats had a higher protein content when compared to annual wheat cultivars, partly due to the low kernel yield of the perennial wheat derivatives. Similarly, several other studies reported that IWG contains more protein than wheat, but exhibits weak gluten-network-forming ability [3,19,20,21]. It is likely that IWG does not contain as high levels of high molecular weight (HMW) glutenins as wheat does, but contains appreciable amounts of gliadins [3].

The Zeleny sedimentation values, dry and wet gluten contents, and gluten index of the control and IWG-supplemented flours are presented in Table 2. There were significant differences bet11ween flour samples in terms of sedimentation values, dry and wet gluten contents, and gluten index (*p* < 0.05). Sedimentation values of 15, 30, 45, and 60% IWG substituted flours were in the range of 34–51 mL, while that of control flour was 62 mL. The sedimentation value of IWG flour was 25 mL (Table 1). Generally, the sedimentation level of wheat flour suitable for bread making should be above 30 mL for acceptable bread quality, which justified our selection of only up to 60% IWG flour incorporation into the formula. Similar results were reported in the literature, indicating that the sedimentation values of perennial wheats are lower compared to those of annual wheat cultivars [18].

The dry (9.9%) and wet (29.2%) gluten contents of the control flour were lower compared to those of the IWG flour (15.4 and 38.8% for dry and wet gluten, respectively) (Table 1). The dry and wet gluten contents of the IWG substituted flours were in the range of 11.1–13.8% and 32.6–39.0%, respectively (Table 2). As expected, the dry and wet gluten contents of the flour samples increased significantly (*p* < 0.05) with the increasing IWG flour substitution level. The gluten index of the control and IWG substituted flours were 98, 87, 63, 50, and 43, in ascending order of IWG substitution (Table 2), while the gluten index of IWG flour was 28 (Table 1). The gluten index of the flour samples generally decreased with the increasing substitution levels of IWG flour. It is evident from Table 2 that replacing a good-gluten-quality wheat flour with a low-gluten-quality flour such as IWG decreases the overall gluten quality, and these findings are in line with those in the literature [22].

The Farinograph results (dough development time, water absorption, stability, softening degree, and Farinograph quality number) of the control and blended flours are presented in Table 3. High water absorption, high stability, and low softening degree in flour are indicators of good bread-making quality. As expected, the Farinograph properties of the control flour were better than those of the flour samples substituted with IWG. The Farinograph dough development time, water absorption, stability, and quality number values of the doughs generally decreased as the level of IWG flour incorporation (15% to 60%) increased. Marti et al. [23] also concluded that IWG enrichment resulted in a decrease in dough development time and dough stability, which is in agreement with the present study. With the addition of IWG flour to wheat flour, the Farinograph stability and the gluten index of flours, which are good indicators of gluten quality, decreased. Marti et al. [11] highlighted the weakening effect of IWG addition on the gluten network and rheological properties.

### 3.2. Breadmaking Quality

Bread volume is one of the main criteria to determine the breadmaking quality of flours. Volumes of the bread samples produced with control and IWG-substituted flours are presented in Table 4. The loaf volume of the control bread was 445 mL, while the loaf volumes of the IWG-substituted breads were in the range of 360–485 mL. Bread produced with the addition of 15% IWG flour had a significantly higher (*p* < 0.05) loaf volume among all the bread samples, including the control bread (Figure 1). Extra-strong flour/dough, *cv*. Cavus, being one of the strongest breadmaking cultivars from Türkiye, may result in loaves with a lower loaf volume and symmetry due to its very stiff gluten structure. Gluten proteins in these dough systems can be modified by reducing agents, diluted with a weaker flour or a flour other than wheat flour to produce a bread with better quality [24,25,26].

A higher amount of IWG flour (more than 15%) addition to wheat flour caused lower bread volumes, probably due to the lower gluten quality of the IWG flour compared to the control flour. The lowest bread volume was observed for the bread made from the highest IWG-containing flour (Figure 1). These results are in line with the findings of Banjade et al. [21], who showed that IWG bread produced using 100% refined IWG flour had a lower volume than the wheat bread produced using 100% hard red winter wheat flour. IWG was found to be deficient in high-molecular-weight glutenins (HMW-GS) [3,23] and rich in low-molecular-weight glutenins (LMW-GS) [20], which are the possible reasons for the lower bread volume of IWG bread compared to wheat flour bread [3,21]. Gluten proteins are connected to breadmaking quality; especially, HMW-GS is thought to be the main predictor of bread-making quality [27]. Despite lower bread volumes at high IWG concentrations (i.e., >30% IWG flour substitution of wheat flour), bread shape results indicated that IWG flour can be used as an ingredient in bread formulas up to 45% substitution of wheat flour. However, at the 60% substitution level, the bread collapsed and the loaf shape was deformed. This was likely due to the inferior gluten quality and thus weak gas holding capacity of the IWG-flour-containing dough, which could not hold in the CO_2_ produced during dough fermentation.

Quality evaluation characteristics of the bread samples are presented in Table 4. Quality characteristics of symmetry, crust colour, crumb cell structure, crumb colour, and softness values were assessed. Albeit statistically insignificantly, bread with 15% IWG flour substitution had slightly better symmetry, crust colour, and softness values when compared to the control bread. Both the control and the bread with 15% IWG substitution had significantly better (*p* < 0.05) symmetry, crust colour, and softness values compared to the breads with higher levels of IWG flour substitution.

### 3.3. Effect of IWG Substitution Levels on Breads Staling

The texture analysis to evaluate the staling properties of the bread samples revealed that crumb firmness values on day 1 were significantly lower (*p* < 0.05) for bread containing 30% and lower amounts of IWG flour compared to bread containing 45 and 60% IWG (Table 5). The crumb firmness values of all breads increased from day 1 to day 3, indicating increased staling, as expected. These results in an agreement with Lazaridou et al. [28], who stated that crumb hardening occurs during the storage of breads, with bread firmness values increasing with increasing storage time. On day 3, among all the breads studied, including the control, crumb firmness was the lowest for the 15% IWG-flour-substituted bread (1046 g-force (g-f)), while it was the highest for the 60% IWG-flour-substituted bread (2149 g-f) (*p* < 0.05). The hardness results of the present study are in line with those of Dai et al. [29], who reported that the hardness values of IWG breads were in the range of 12.4–15.1 N (~1265–1540 g-f). The addition of a low-quality-gluten-containing flour can reduce the elasticity of dough [30] and affect the hardness of breads made from it [31].

### 3.4. Yellow Pigment Content and Colour Values of IWG-Substituted Breads

The yellow pigment contents of the bread samples are presented in Table 6. The yellow pigment contents of bread samples significantly increased with the increasing IWG substitution levels (*p* < 0.05), due to the very high yellow pigment content of IWG flour (23.09 µg/g, Table 1). There are no data about the carotenoid and yellow pigment contents of IWG-substituted breads in the literature; however, the content of yellow pigment and carotenoids in IWG grains and ancient wheat varieties were reported previously. For example, Kaplan Evlice [32] reported the yellow pigment content of einkorn whole flour and durum wheat flour to be 12.1 and 7.7 µg/g, respectively. Tyl and Ismail [3] concluded that most IWG genotypes had much greater levels of two main carotenoids, lutein and zeaxanthin than wheat. Gazza et al. [18] stated that perennial wheat genotypes had higher amounts of carotenoids compared to their annual counterparts. Given that IWG flour has a substantially higher yellow pigment content compared to durum wheat (a wheat species that is already known for its higher yellow pigment content relative to bread wheat) and ancient wheat flours, the yellow pigment content of breads that contain IWG flour would likely be much higher compared to breads that contain durum wheat and ancient wheat flours. These results point to the potential health benefits of IWG incorporation into bread.

Colour is an important characteristic of food products that affects food’s acceptability and consumer perception. Flour is one of the main factors that determine bread colour [33]. Crumb and crust L*, a*, and b* colour values of the bread samples are presented in Table 6. The L*, a*, and b* colour values of the IWG flour were 96.50, 1.46, and 18.58 (Table 1). Generally, there were significant differences between the bread samples in terms of all colour values (*p* < 0.05). The L* values of the bread crumb generally decreased while a* and b* values significantly increased with the increasing IWG flour incorporation levels (*p* < 0.05). The a* and b* colour values of the crumb were the highest for 60% IWG-flour-substituted bread. Among all the breads studied, the 60% IWG-flour-substituted bread resulted the brightest crust colour. There were no major changes in crust colour to affect consumer acceptability.

### 3.5. Total Phenolic Content (TPC) and Antioxidant Activities of Bread Samples Enriched with IWG Flour

The TPC and antioxidant activity results of the control bread and the breads substituted with IWG flour are presented in Table 7. The TPC of the control bread sample was determined to be 20.61 mg GAE/100 g bread, while the TPCs of the breads substituted with IWG flour were substantially higher and between 21.77 and 32.03 mg GAE/100 g bread. The TPC values of the IWG-substituted breads significantly increased as the ratio of IWG percentage in the bread formulation increased (*p* < 0.05).

The DPPH values of the breads substituted with IWG flour ranged from 8.68 to 19.61 mg TE/100 g bread, whereas the DPPH value of the control bread was 6.81 mg TE/100 g (Table 7). The ABTS values of the breads substituted with IWG flour were between 54.49 and 88.78 mg TE/100 g bread, whereas the ABTS value of the control bread was 42.13 mg TE/100 g. The DPPH and ABTS scavenging activity values of the breads also significantly increased (*p* < 0.05) with the increase in the IWG flour substitution level in bread formulation (Table 7). There are no previous reports in the literature on the phenolic content of breads produced using IWG flour. However, phenolic acid profiles of IWG-insoluble fibre were previously reported. It was concluded that the concentration (6750 µg/g corrected insoluble fibre) of trans-ferulic acid in IWG-insoluble fibre was comparable to that of insoluble wheat and rye fibres. Furthermore, IWG generally contained higher levels of antioxidants than bread wheat [3,34]. Three hydroxycinnamic acids; ferulic, p-coumaric, and sinapic acids, were also identified and quantified previously. Tyl and Ismail [3] stated that hard red wheat had significantly lower p-coumaric acid than all IWG cultivars studied, but ferulic and sinapic acid contents were similar for wheat and IWG cultivars. On the other hand, Gazza et al. [18] stated that there was not a clear-cut difference between perennial wheat genotypes and wheat varieties in terms of soluble polyphenol contents.

## 4. Conclusions

This study highlighted some quality characteristics and bread-making quality of *Thinopyrum intermedium* as a novel ingredient in bakery applications and compared them with common bread wheat characteristics. Overall, IWG-containing flours had significantly higher dry gluten and wet gluten values, but lower Zeleny sedimentation values. Doughs made from IWG-containing flours had lower Farinograph water absorption, stability, and quality number values and higher softening degree, indicating inferior dough rheological properties with the IWG addition to wheat flour. The bread quality analysis revealed that 15% IWG-containing bread had the highest bread volume (*p* < 0.05) while performing as good as the control bread. When the staling properties of the breads were compared, the 15% IWG-containing bread had the lowest bread firmness value on day 3 (*p* < 0.05), indicating the lowest rate of staling at this IWG substitution level. Overall, all IWG-containing breads had significantly higher yellow pigments and total phenolic contents as well as higher antioxidant activity when compared to the control bread, pointing to the potential health benefits of this novel ingredient. Future work will focus on different food uses for whole IWG flour and characteristics of the food products made from it.

## Figures and Tables

**Figure 1 foods-12-02109-f001:**
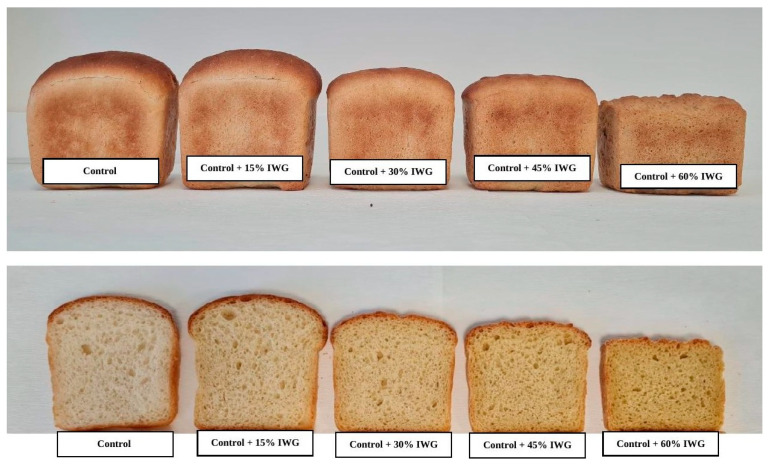
Control bread and IWG-flour-supplemented bread samples.

**Table 1 foods-12-02109-t001:** IWG and control flour properties.

Properties		IWG Flour	Control Flour
Protein (%, dmb)		19.3 ± 0.32	12.0 ± 0.00
Zeleny sedimentation value (ml)		25 ± 0.0	62 ± 0.7
Dry gluten (%)		15.4 ± 1.13	9.9 ± 0.07
Wet gluten (%)		38.8 ± 0.35	29.2 ± 0.49
Gluten index		28 ± 4.2	98 ± 0.2
Colour	L*	96.5 ± 0.30	99.4 ± 0.08
	a*	1.46 ± 0.007	1.41 ± 0.057
	b*	18.58 ± 0.375	11.02 ± 0.071
Yellow pigment content (µg/g)		23.09 ± 0.085	2.78 ± 0.021

**Table 2 foods-12-02109-t002:** Zeleny sedimentation, dry and wet gluten values, and gluten index of control flour and flours supplemented with IWG flour.

Sample	Zeleny Sedimentation Value (mL)		Dry Gluten (%)		Wet Gluten (%)		Gluten Index	
Control	62 ± 0.7	a	9.9 ± 0.07	e	29.2 ± 0.49	e	98 ± 0.2	a
15% IWG	51 ± 0.7	b	11.1 ± 0.07	d	32.6 ± 0.07	d	87 ± 6.9	b
30% IWG	43 ± 0.0	c	11.5 ± 0.07	c	35.2 ± 0.49	c	63 ± 4.9	c
45% IWG	38 ± 0.0	d	12.5 ± 0.21	b	36.6 ± 0.28	b	50 ± 2.8	d
60% IWG	34 ± 0.4	e	13.8 ± 0.00	a	39.0 ± 0.49	a	43 ± 1.3	d

Values followed by different letters in the same column are significantly different (*p* < 0.05).

**Table 3 foods-12-02109-t003:** Farinograph results of flours supplemented with IWG flour.

Sample	Dough Development Time (min)		Water Absorption (%)		Stability (min)		Softening Degree * (BU)		Quality Number	
Control	7.47 ± 0.297	b	62.5 ± 0.57	a	15.40 ± 1.414	a	40 ± 9.9	d	170 ± 7.1	a
15% IWG	8.12 ± 0.141	a	61.9 ± 0.71	a	10.66 ± 0.559	b	79 ± 12.0	c	143 ± 9.9	a
30% IWG	5.92 ± 0.283	c	59.9 ± 0.64	b	8.22 ± 0.318	c	111 ± 13.4	b	109 ± 14.1	b
45% IWG	5.40 ± 0.170	c	57.4 ± 0.64	c	6.68 ± 0.311	cd	143 ± 5.7	a	87 ± 7.1	bc
60% IWG	3.75 ± 0.141	d	56.6 ± 0.90	c	5.31 ± 0.085	d	159 ± 9.9	a	69 ± 12.7	c

Values followed by different letters in the same column are significantly different (*p* < 0.05). * 12 min after the development time.

**Table 4 foods-12-02109-t004:** Volumes and quality evaluation of bread samples supplemented with IWG flour.

	Bread Volume (mL)		Quality Evaluation									
			Symmetry		Crust Colour		Crumb Cell Structure		Crumb Colour		Softness	
Control	445 ± 7.1	b	4.8 ± 0.35	a	4.8 ± 0.35	a	9.5 ± 0.71	a	10.0 ± 0.00	a	9.3 ± 0.35	a
15% IWG	485 ± 0.0	a	5.0 ± 0.00	a	5.0 ± 0.00	a	8.3 ± 0.35	a	9.5 ± 0.71	ab	9.8 ± 0.35	a
30% IWG	423 ± 3.5	c	3.5 ± 0.00	b	5.0 ± 0.00	a	6.3 ± 0.35	b	9.0 ± 0.00	b	7.8 ± 0.35	b
45% IWG	395 ± 7.1	d	2.3 ± 0.35	c	3.8 ± 0.35	b	4.5 ± 0.71	c	7.8 ± 0.35	c	6.3 ± 0.35	c
60% IWG	360 ± 14.1	e	1.3 ± 0.35	d	2.3 ± 0.35	c	3.5 ± 0.71	c	6.8 ± 0.35	d	3.5 ± 0.71	d

Values followed by different letters in the same column are significantly different (*p* < 0.05).

**Table 5 foods-12-02109-t005:** Staling properties of bread samples supplemented with IWG flour.

Samples	Firmness (g-f)			
	Day 1		Day 3	
Control	674 ± 67.8	cd	1436 ± 49.1	b
15% IWG	654 ± 24.8	d	1046 ± 76.7	c
30% IWG	777 ± 23.0	c	1587 ± 269.6	b
45% IWG	1103 ± 45.0	b	1577 ± 115.9	b
60% IWG	1365 ± 34.0	a	2149 ± 80.7	a

Values followed by different letters in the same column are significantly different (*p* < 0.05).

**Table 6 foods-12-02109-t006:** Yellow pigment content and crust and crumb colour values of breads supplemented with IWG flour.

Samples	Yellow Pigment Content (µg/g)		Crust Colour of the Bread						Crumb Colour of the Bread					
			L*		a*		b*		L*		a*		b*	
Control	2.66 ± 0.021	e	42.7 ± 2.16	b	17.9 ± 0.48	a	31.0 ± 2.07	b	78.0 ± 0.68	a	0.5 ± 0.17	e	21.2 ± 0.47	e
15% IWG	4.67 ± 0.000	d	40.3 ± 0.44	bc	18.1 ± 0.16	a	27.5 ± 0.16	c	70.7 ± 6.44	b	1.5 ± 0.08	d	26.0 ± 0.39	d
30% IWG	5.00 ± 0.000	c	39.1 ± 2.02	c	18.0 ± 0.50	a	26.6 ± 2.07	c	70.1 ± 1.06	bc	2.5 ± 0.12	c	29.7 ± 0.18	c
45% IWG	6.38 ± 0.085	b	40.7 ± 0.72	bc	17.4 ± 0.43	a	27.7 ± 2.64	c	67.8 ± 1.08	bc	3.6 ± 0.21	b	33.1 ± 0.47	b
60% IWG	6.98 ± 0.043	a	48.9 ± 0.15	a	14.5 ± 0.69	b	37.1 ± 0.17	a	64.9 ± 0.57	c	4.3 ± 0.09	a	34.4 ± 0.20	a

Values followed by different letters in the same column are significantly different (*p* < 0.05).

**Table 7 foods-12-02109-t007:** Total phenolic content and antioxidant activities of breads supplemented with IWG flour.

Sample	TPC mg GAE/100 g Bread		DPPH mg TE/100 g Bread		ABTS mg TE/100 g Bread	
Control	20.61 ± 0.10	e	6.81 ± 0.31	c	42.13 ± 0.70	e
15% IWG	21.77 ± 0.35	d	8.68 ± 0.94	c	54.49 ± 0.42	d
30% IWG	26.22 ± 0.20	c	12.74 ± 0.62	b	61.24 ± 0.14	c
45% IWG	27.37 ± 0.15	b	14.92 ± 0.94	b	76.83 ± 0.28	b
60% IWG	32.03 ± 0.30	a	19.61 ± 1.25	a	88.78 ± 0.14	a

Values followed by different letters in the same column are significantly different (*p* < 0.05).

## Data Availability

The data presented in this study are available on request from the corresponding author.

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
