# Peer review of "Utilization of Intermediate Wheatgrass (Thinopyrum intermedium) as an Innovative Ingredient in Bread Making"

_foods, 2023, doi:10.3390/foods12112109_

Round 1

Reviewer 1 Report

The manuscript aims to asses quality of bread produced using Thi-nopyrum intermedium flour. The manuscript is well prepared and easy to understand by vast spectrum of readers. The introduction highlights the topic properly, however the novelty of the manuscript could be more emphasized. The method section is clearly explained. The discussion is simple, but properly lead and with relevant discussion with literature. The manuscript contain 31 sources that have been published mostly within last 5 years.

Detailed remarks:

It is not clear why sensory evolution was based on hedonic scale?

The sample coding should be changes, please remove control form IGW samples

Table 1 pleas provide control flour properties

Table 3 lacks SD

Reference list is not edited according to journal requirements

Reviewer 2 Report

The objective of the study is clear and worthy of study. The manuscript subject is very interesting However, some modifications need to be addressed.

- abbreviations should be spilled out in their first mention

- In the introduction part should be more highlighted the main aim of the paper.

- the quality of the figures needs to be enhanced, and the use of contrasting colors will be more obvious. e.g. Figure.1

-I could not see some innovative information in this piece of research.

-Authors have no regard for punctuation. Many improper uses of punctuation have been found.

-Some paragraphs need to be broken for easy understanding.

-Abstract lacks data values based on the study's findings relating to different parameters.

-Scientific name must be written in italic.

-The method should be summarized and made it more clear.

-Results of the study have not been compared with the findings of earlier scientists.

-Add conclusion and future perspective in one to two lines.

-Replace the references of older years with the reference of latest years. 

Reviewer 3 Report

The utilization of intermediate wheatgrass (Thinopyrum intermedium) in bread making is evaluated in this study. The research is interesting and practical. But it needs some revisions:

1) Other studies about the utilization of IWG in bread making should be added.

2) The information about the starch, mineral, lipid contents of IWG should be added.

3) The statistic analysis in Table 3 should be added.

4) Please check the result of total phenol content determination. The result is implausible:  The TPC of the control bread sample was determined as 20.61 g GAE/100g bread.

5) This expression is also not credible, “It was concluded that the concentration (6750 mg/g corrected insoluble fiber) of trans-ferulic acid in IWG insoluble fiber ”.

Reviewer 4 Report

Totally, the research has been well designed, although, there are some points to review as follows:

The proof editing of English language is required for all the text. There are some errors and mistakes present in the text.

section 2.2.2: Farinograph properties of the "flours" should be replaced to "dough" 

Table 3: please add statistical analysis to indicate significant different among means in the table

Sensory anlysis: 3 panelists are not suffient for evaluating the sensory properties. Increase the number of panelist to at least 15 persons

Section 2.2.3. The references should be updated, use below reference for updating the references in this section.

Christodoulou, M.C.; Orellana Palacios, J.C.; Hesami, G.; Jafarzadeh, S.; Lorenzo, J.M.; Domínguez, R.; Moreno, A.; Hadidi, M. Spectrophotometric Methods for Measurement of Antioxidant Activity in Food and Pharmaceuticals. Antioxidants 202211, 2213. https://doi.org/10.3390/antiox11112213

Round 2

Reviewer 4 Report

The last comment haven't done correctly! please do it completely based on reviewer's comment!
